

# Antarctic grounding zone characteristics from CryoSat-2

Geoffrey J. Dawson[1] and Jonathan L. Bamber[1]

[1]Bristol Glaciology Centre, School of Geographical Sciences, University of Bristol, Bristol, UK

**Correspondence:** Geoffrey J. Dawson (geoffrey.dawson@bristol.ac.uk)

**Abstract.** We present the results of mapping the limit of tidal flexure (point F) and hydrostatic equilibrium (point H) of the grounding zone of Antarctic ice shelves from CryoSat-2 standard and swath elevation data. Overall we were able to map 41 % of the grounding zone of the larger floating ice shelves and outlet glaciers in Antarctica. We obtain near-complete coverage of the Filchner-Ronne Ice Shelf and partial coverage of the Ross Ice Shelf, Dronning Maud land and the Antarctic Peninsula, while we could not map a continuous grounding zone for the Amery Ice Shelf and the Amundsen Sea Sector. Tidal amplitude and distance south (i.e. across track spacing) are controlling factors in the quality of the coverage and performance of the approach. The location of the point F agrees well with previous observations that used differential satellite radar interferometry (DInSAR) and ICESat-1, with an average landward bias of 0.1 km and 0.6 km and standard deviation of 1.1 km and 1.5 km for DInSAR and ICESat measurements, respectively. We also compared the results directly with DInSAR interferograms from the Sentinel-1 satellites, acquired over the Evans Ice Stream and the Carlson Inlet (Ronne Ice Shelf) and found good agreement with the mapped points F and H. We present the results of the spatial distribution of the grounding zone width (the distance between points F and H), and used a simple elastic beam model to investigate the relationship between ice thickness and grounding zone width.

## 1 Introduction

In Antarctica, the majority of the grounded ice sheet (74 %, *Bindschadler et al.* (2011)) abuts floating ice shelves or outlet glaciers. It is in this grounding zone where the ocean can directly influence the inland ice sheet. The grounding zone delineates the different stress regimes of grounded and freely floating ice. Grounded ice that was once supported by the bed is transitioning to thefreely floating ice shelf, and is supported partially by internal stresses and by hydrostatic pressure. The precise point at which the ice sheet detaches from the bed (i.e. the grounding line) may vary on short time scales modulated by tidal motion and bedrock slope. Ice thickness, basal drag and side drag may also vary across the grounding zone, causing rapid changes in ice velocity. Understanding ice dynamics and structure across the grounding zone is important for mass budget calculations and makes it a critical boundary for ice sheet modelling. In areas of low bedrock slope, changes in ice thickness in the grounding zone can lead to large horizontal changes grounding line location. For example grounding line retreat in the Amundsen Sea Sector (*Rignot et al.*, 2014; *Christie et al.*, 2016; *Scheuchl et al.*, 2016), caused by dynamic thinning of this part of the ice sheet (*Shepherd et al.*, 2002; *McMillan et al.*, 2014), has highlighted the need to monitor changes in grounding zone location as one measure of of ice sheet stability.



It is not possible to remotely measure the actual location of the grounding line, instead, we can study ice shelf flexure or surface geometry (such as break in slope) to infer its position. The inner limit of tide-induced ice sheet flexure (point F), is commonly used as a proxy for the grounding line. Point F can be mapped using differential satellite radar interferometry

(DInSAR) (*Gray et al.*, 2002; *Rignot*, 1998b), repeat track analysis of ICESat (Ice, Cloud, and land Elevation Satellite) laser altimetry (*Fricker and Padman*, 2006) and CryoSat-2 radar altimetry (*Dawson and Bamber*, 2017). We can also identify the point past which the ice shelf is in hydrostatic equilibrium (point H), providing a measure of the width of the grounding zone (i.e. the distance between points F and H, *Fricker and Padman* (2006)). However, currently DInSAR or ICESat techniques do not have sufficient spatial or temporal coverage to monitor change across the entire grounding zone. Break-in-slope methods

(*Bohlander and Scambos*, 2007; *Bindschadler et al.*, 2011; *Bamber and Bentley*, 1994; *Hogg et al.*, 2017) between the flat ice shelf and the grounded ice sheet can also allow us to map the grounding line and monitor retreat, but in regions where there is not a clear break in slope, this technique can be unreliable or ambiguous (*Bamber and Bentley*, 1994; *Fricker and Padman*, 2006; *Brunt et al.*, 2010; *Rignot et al.*, 2011; *Depoorter et al.*, 2013).

We can also use the tidal flexure of the ice sheet to investigate the structure of the grounding zone. This can help us charac-

terise stress gradients and ice rheology. *Holdsworth* (1977) first used an elastic beam model as an analogue for the grounding zone. This enabled studies that used tilt-meters (*Stephenson*, 1984) and kinematic GPS methods (*Vaughan*, 1995) to measure the tidally induces deformation across the grounding zone and determine the elastic (Young's modulus) properties of the ice. More recently, DInSAR was used remotely to measure the magnitude of the tidally induces deformation across the grounding zone (*Rabus and Lang*, 2002; *Sykes et al.*, 2002), and was combined with numerical elastic models (*Schmeltz et al.*, 2002;

*Marsh et al.*, 2014) to estimate ice thickness distributions and ice properties across the grounding zone. These studies have shown that the measured Young's modulus differs substantially from laboratory measurements. Fracturing in the ice can reduce its effective thickness (*Hulbe et al.*, 2016; *Rosier et al.*, 2017), and the elastic modulus can vary through changes in temperature and ice fabric. The ice also does not behave purely elastically over the timescales of tidal motion, and this can be investigated by treating it as a viscoelastic material (*Wild et al.*, 2018).

Studies that have investigated the structure of the grounding zone through tidal flexure have, to date, focused on individual ice streams. The method presented here uses CryoSat-2 radar altimetry to provide a new tool that allows us to map a large fraction of the grounding zone, while also investigating its structure. In this paper, we first present the results of using 7.5 years of CryoSat-2 data to map the Antarctic grounding zone. The results are then validated against previous DInSAR and ICESat measurements. Finally, the grounding line width, $W$ (the distance between point F and H) is then used to investigate

the physical structure of the grounding zone.

## 2  CryoSat-2 Data

CryoSat-2, launched in 2010, uses a synthetic aperture radar interferometric (SARIn) mode near the margins of the ice sheet. This new mode mostly overcomes issues of off-ranging and "loss-of-lock" near breaks in slope which have limited the coverage of conventional satellite radar altimetry over sloping terrain (*Bamber et al.*, 2009). The SARIn mode combines "delay-Doppler"





processing to improve along-track resolution (*Raney*, 1998), with dual antennas to provide the location of the return echo in the cross-track direction (*Jensen*, 1999). This enables the acquisition of elevation measurements based on the first return (point of closest approach or POCA) and "swath processed" heights derived from the time-delayed waveform beyond the first return (*Gray et al.*, 2013). In this study, we used CryoSat-2 POCA and swath elevation data, to measure elevation change due to tidal flexure of the floating ice shelf. These data were derived from the CryoSat-2 SARIn baseline C level 1b product,

with revised star tracker measurements provided by the European Space Agency (ESA). We processed POCA data using the scheme described in *Helm et al.* (2014) which employs a threshold re-tracker, this is less sensitive to any changes in the extinction coefficient of the snow and minimises any potential biases in elevation data. We used a processing scheme that closely follows *Gray et al.* (2013) to process the swath data, and used minimum coherence and power thresholds of 0.8 and -160 db respectively.

The coverage of POCA and swath data is shown in Figure 1 and Figure A1. POCA data provides consistent sampling over flat terrain, such as ice shelves with higher data density at high latitudes due to the narrower track spacing of the satellite. However, as they are based on the first return of the waveform, over sloping terrain, they only provide elevation measurements upslope of satellite nadir. This reduces coverage, particularly near a break in slope, such in the vicinity of the grounding line. Swath data provides elevation estimates downslope of POCA, and to obtain the best coverage of the grounding zone, we need

to use a combination of POCA and swath data. While swath data tends to be noisier (*Gray et al.*, 2017), they have an order of magnitude higher spatial sampling than POCA data. We obtain the highest sampling of swath data near breaks in slope, and over moderately sloping terrain. In high sloping regions with complex topography, we generally lose coverage, for example the Transantarctic Mountains and parts of the Antarctic Peninsula. In these regions, steep slopes can cause the satellite to lose "lock" wherein the return echo of the radar wave is not captured within the range window. Also in areas of complex topography,

there may be more than one point where the radar wave reflects off the ground for a given range, leading to a loss of coherence and an ambiguous location for the located echo in the cross-track direction.

## 3   Methods

Our approach used CryoSat-2 surface elevation measurements to determine the limit of tidal flexure of the ice (F) and the limit of hydrostatic equilibrium (H), and closely followed the technique described in *Dawson and Bamber* (2017). The key feature

of this approach is to use the pseudo-crossover method of *Wouters et al.* (2015), to simultaneously solve for topography, a dimensionless tidal amplitude ($T_d$) and a linear surface elevation rate ($\dot{h}$) using equation (1).

$$h(x, y, p) = a_0 + a_1.x + a_2.y + T_d.p + \dot{h}t \tag{1}$$

Where $h$ is the elevation, $t$ is time, $a_0$ is the mean elevation, $a_1$ and $a_2$ are the slopes of the topography in the $x$ and $y$ direction respectively. We used a model tidal amplitude, p, (the CAT2008a tide model, which is an update to the model described by *Padman et al.* (2002)) calculated at a constant distance of 10 km from the nominal grounding line in *Depoorter et al.* (2013)

to scale $T_d$. Thus, $T_d$ gives a measure of the tidal contribution to the elevation, and $\dot{h}$ measures elevation change not associated with tidal motion e.g. from ice sheet thinning or changes in firn compactions rate of the floating ice.



**Figure 1.** Data coverage plot for POCA and swath data. Red, green and blue data points correspond to where we used POCA, swath or both to calculate the tidal amplitude, $T_d$, respectively. The colour scales show the data density (POCA points/km). The swath data density is scaled by 150 (the average number of swath to POCA data points) to match the POCA density.





We calculated $T_d$ (Figure 2) and $\dot{h}$ (Figure A2) within a 2×2 km grid cell using CryoSat-2 data between 2010 and 2017. We used a 3-year moving window, weighted by a tri-cube weight function, resulting in 6 yearly measurements for each grid cell between 2011 and 2017. By using a 3-year moving window, we ensured that there were at least 4 different satellite passes per grid cell while allowing for any dynamic changes in elevation of the ice sheet that may have occurred. An alternative method would be to calculate temporal changes in $T_d$ and $\dot{h}$ over the entire time series. This would require including additional parameters, however, which may risk over-fitting the data. We then calculated the mean of $T_d$, to obtain a single value over the observation period. We only used data where $-0.5 < T_d < 1.5$ and $|T_d - \tilde{T}_d| < 0.5$ where $\tilde{T}_d$ is the median values of the yearly measurements per cell. This removed any poor fits to equation 1, which likely come from erroneous elevation data. This method could potentially monitor grounding line retreat, for example, in the Amundsen Sea Sector (*Rignot et al.*, 2014; *Christie et al.*, 2016; *Scheuchl et al.*, 2016). However, as these areas have a small range of tidal amplitude and poor data coverage (see Section 4.1), we could not detect a significant change during the observation period (2010-2017).

When ice is in hydrostatic equilibrium, the real tidal amplitudes match the closest model tidal amplitudes, and we find $T_d = 1.0 \pm 0.2$. Over grounded ice, we find $T_d = 0.0 \pm 0.2$, as there is no correlation between elevation and model tidal amplitudes. Previously we mapped the point F by considering ice to be influenced by the vertical motion of the tides above a certain threshold. This introduced a seaward bias, as we did not resolve amplitudes below the threshold value. In this study, we fitted an error function perpendicular to the grounding zone to determine $F$ and $H$, and removed any potential bias. This process was performed iteratively: We first mapped the centre line of the grounding zone (i.e. $T_d = 0.5$ contour) with a 1000 m spacing. We then sampled $T_d$ perpendicular to the initial guess of the centre line of the grounding zone, and fitted an error function to find a new location for $T_d = 0.5$ as well as points F ($T_d = 0.1$) and H ($T_d = 0.9$). The centre line of the grounding zone was then re-sampled to 1000 m spacing, and the process repeated. The process was repeated at least three times or until the grounding line location did not change significantly by visual inspection. To make the fitting method more robust, we fixed the maximum and minimum value to 1 and 0, respectively, and weighted the fitting process around $T_d = 0.5$, using a tri-cube weight function. We only included data points where the grounding line width was calculated between 100 m and 10000 m, and where there was continuous coverage of $T_d$ across the grounding zone. We split the grounding line when there was a break greater than 4 km, and removed any segments of mapped grounding line shorter than 20 km. Finally, we applied a 10 km along-track smoothing using the Polynomial Approximation with Exponential Kernel (PAEK) smoothing algorithm.

## 4 Grounding zone mapping

### 4.1 Coverage

We were able to map the grounding zone (points F and H) for 41 % of the main floating ice shelves or outlet glaciers. The percentage mapped for several key regions are shown in Table 1. In the high latitude areas of the Ross Ice Shelf and the Filchner-Ronne Ice Shelf, we obtained near complete coverage. In these regions, the track spacing of CryoSat-2 is as low as 0.5 km resulting in a high spatial sampling of the grounding zone, and we only lost coverage in high sloping regions with complex topography, for example, the Transantarctic Mountains.





At lower latitudes, further north the coverage is variable. The spatial sampling is lower as the track spacing of the satellite varies from 2 km to 3 km. In the high sloping, low tidal range (0.8 m - 1 m) Amundsen Sea sector and the Amery Ice Shelf, we could not map a continuous grounding line. There were very few POCA data near the grounding zone, and the swath data were too noisy to resolve the tidal signal. Also, over fast-flowing ice shelves such as Pine Island and Thwaites glacier, any surface features such as ridges will move along the direction of flow. As the surface is not sampled at the same time, this will

result in a spread of elevation measurements over these features, introducing noise. Using a Lagrangian framework to correct for the movement of the ice shelves, is an effective way of removing this source of noise (*Moholdt et al.*, 2014). However, a Lagrangian framework cannot be used in this study, as it is only valid on floating ice shelves and not over grounded ice or the grounding zone.

     The coastline of Dronning Maud Land is also at relatively low latitudes, however the tidal range is higher (1 m - 2 m) and

we were able to resolve the tidal signal using primarily swath data. Here, we obtained 41 % coverage of the grounding zone for Dronning Maud Land. In the lower latitude areas of the Antarctic Peninsula, the track spacing of the satellite ranges from 3 km to 4 km. The spatial sampling of both POCA and swath data is lower, and we were able to map 11 % of the grounding zone. To improve the coverage in low latitude areas we could have increased the cell size from 2 km, however we would then lose the spatial resolution needed to map the grounding zone accurately.

## 4.2    Validation with DInSAR and ICESat observations

We first compared point F mapped using CryoSat-2 to the previous mapping methods that used DInSAR observations (*Rignot et al.*, 2016; *ESA Antarctic Ice Sheets CCI*, 2017) and ICESat (*Brunt et al.*, 2010) repeat-track analysis. The absolute distance (or bias) and the standard deviation between the CryoSat-2 grounding line (defined as point F here) and the DInSAR/ICESat grounding lines for several regions are shown in Table 1. Across the whole of Antarctica, the absolute distance between the

DInSAR and ICESat groundings lines and the CryoSat-2 grounding line is -0.1 km (a negative value represents a landward bias) for both datasets, showing that there is a landward bias between the CryoSat-2 method and others, which does not change significantly with region. The standard deviation is 1.1 km and 1.5 km between the DInSAR and ICESat groundings lines and the CryoSat-2 grounding line, respectively, however, this varies with region. In the high latitude areas of the Ross and Filchner-Ronne Ice Shelves, the standard deviation is low (1.7 km between the CryoSat-2 and DInSAR grounding lines), and

the grounding line matches well. While in the lower latitude areas with large tidal range (Dronning Maud land and Amery Ice Shelf), there is a standard deviation of 2.0 km between the CryoSat-2 and DInSAR grounding lines. This increase in standard deviation is due to reduced data density at lower latitudes, and also the smaller tidal range, which results in a noisier calculation of $T_d$ and a larger deviation from previous observations. In comparison, the grounding line mapped using *Rignot et al.* (2016) has a bias of -0.3 km with a standard deviation of 0.9 km when it is compared to the *ESA Antarctic Ice Sheets CCI* (2017)

grounding line and a bias of -0.4 km with a standard deviation of 1.1 km when compared to the ICESat grounding line.

     We also compared our results directly with DInSAR interferograms from the Sentinel-1 satellites, acquired over the Evans Ice Stream and the Carlson Inlet (Ronne Ice Shelf). We used single look complex (SLC) SAR images acquired by the Sentinel-1 satellites in the interferograms wide swath mode. The SAR operates in the C-band at 5.405 GHz, and in the wide swath





**Table 1.** The percentage of grounding line mapped along with the bias (a negative value represents a landward bias) and standard deviation between the CryoSat-2 mapped grounding line (Point F) and the DInSAR and ICESat mapped groundings lines, for several regions across Antarctica (shown in Figure 1)

| Area | | Bias (km) | | | Standard deviation (km) | | |
|---|---|---|---|---|---|---|---|
| | % mapped | DInSAR (M) | DInSAR(E) | ICESat | DInSAR (M) | DInSAR (E) | ICESat |
| Antarctica | 32 | - 0.1 | -0.1 | - 0.6 | 1.1 | 1.2 | 1.5 |
| Filchner-Ronne Ice Shelf | 90 | -0.1 | 0.1 | - 0.6 | 1.1 | 1.2 | 1.2 |
| Ross Ice Shelf | 43 | -0.1 | -0.1 | - 0.6 | 0.9 | 1.0 | 1.5 |
| Dronning Maud Land | 41 | 0.1 | 0.3 | -0.9 | 1.2 | 1.3 | 1.9 |
| Antarctic Peninsula | 11 | -0.1 | -0.1 | –0.6 | 1.3 | 1.2 | 1.9 |

mode, lead to a 5x20 m resolution in ground range and azimuth. Each satellite has a repeat cycle of 12 days, and by using both
Sentinel-1A and 1B, we were able to form the double-differenced interferograms from 3 scenes spanning between 21st July
and 3rd August 2018, and data were processed using GMTSAR. By calculating the difference between two interferograms, we
removed any signal that is common among both interferograms, (e.g. constant ice flow) and only measured changes in ice flow
and deformation of the ice sheet. This region of the Filchner-Ronne Ice Shelf is a relatively stable area, and over the time frame
of measurement, any elevation change will likely be due to tidal deformation. This results in very little measured deformation
over grounded ice and a series of interference fringes that corresponds to the change in height between the two interferograms
due to tides. The landward and seaward limit of these fringes can be robustly interpreted as point F and H, respectively.

The double difference interferogram is shown in Figure 3, and the inner and outer limit of interference fringes which correspond to the boundaries of the grounding zone agree well with the mapped points F and H from CryoSat-2. Each fringe
corresponds to approximately 2.8 cm change in height, and by unwrapping the interferogram using the snaphu method (*Chen
and Zebker*, 2001), we were also able to compare the difference in height caused by tidal deformation. Three cross-sections
over the Evans Ice Stream and the Carlson Inlet are shown in Figure 4, and by normalising the deformation, we could compare
the results directly for two cross-sections (the third cross-section did not have any usable SAR data). In both cross-sections $T_d$
approximately matches the deformation measured by DInSAR and points F and H match well. However, $T_d$ does not match
the exact shape of the deformation. In the DInSAR data, we observe a sharp transition between fully grounded and partially
grounded ice and a smoother transition to fully floating ice. This detail is not captured by CryoSat-2 as it does not have the
precision to detect these small changes in elevation to resolve the tidal deformation fully.

## 5   Grounding zone structure

The width of the grounding zone ($X$), is shown in Figure 5 for several regions across Antarctica. $X$ ranges between 0.5 km and
10 km and has a strong regional variation. The widest grounding zones were found over the Mercer (Ross Ice Shelf), Institute
and Möller Ice Streams (Filchner Ice Shelf). While the narrowest regions were found over ice shelves of Dronning Maud Land.



To a first approximation, this variation in grounding line width can be attributed to the ice thickness in the grounding zone (see Figure 5). The thicker ice tends to be more inflexible, and consequently the internal stresses of the ice can support the ice further from point F. We can demonstrate this by modelling the grounding zone as a semi-infinite beam of constant thickness (*Holdsworth*, 1977). With this model, the vertical deflection of the beam ($w$) is described by

$$w(x) = A_0[e^{-\beta x}(\cos \beta x + \sin \beta x)] \tag{2}$$

where the beam is pinned at a hinge line at x=0 and it is displaced vertically by $A_0$. The spatial wavenumber, $\beta$, is given by

$$\beta^4 = 3\rho_w g \frac{1 - \mu^2}{Eh^3} \tag{3}$$

where $h$ is the ice thickness, $E$ the Young's modulus, $\mu$ the Poisson ratio, $\rho_w$ the density of seawater and $g$ the acceleration due to gravity. Given this relationship, the strongest dependence on spatial wavenumber is the thickness of the ice. *Bindschadler et al.* (2011) used this relationship, the elastic properties of ice ($\mu = 0.3$ and $E = 0.88\,GPa$) and parameters from the Rutherford ice stream (*Vaughan*, 1995) to estimate the grounding line width, $X = (22.2 \pm 6.2)h^{3/4}$. If we compare $X$ to ice

shelf thickness measurements (*Chuter and Bamber*, 2015) at point H calculated from CryoSat-2 POCA elevation data using the assumption of hydrostatic equilibrium, we find $X = (26.4 \pm 6)h^{3/4}$, which agrees well with the previous relationship (Figure 6).

There is considerable scatter between these results due to significant measurement error from both $h$ and $X$ and because other factors, such as ice rheology, vary regionally. Ice shelf thickness measurements have shown to have a mean percentage

error of $4.7\%$ near the grounding zone of the Amery Ice shelf (compared to radio echo sounding measurements (*Chuter and Bamber*, 2015). These errors could be larger in some areas, due, for example, to uncertainties in firn compaction in areas of compressive flow (*Bamber and Bentley*, 1994) and varitions in damage mechanics along shear margins. These parameters may vary over areas where there are rapid changes in ice dynamics and bed topography, such as the grounding zone, potentially introducing larger errors. The grounding zone width is also dependent on ice rheology, the motion of the ice sheet, grounding

line geometry and tidal range and all these factors will also contribute to the observed scatter.

The effect of grounding zone shape can be seen if we look at two cross-sections over the Evans Ice Steam (profiles B and C in Figures 3 and 4). Profile B is situated over a concave section of grounding zone, while profile C is over a straight section of grounding zone. The ice thickness at point H is 868 m and 717 m, and this corresponds to an estimated grounding zone width of 3550 m and 3075 m, using the simple model of the grounding zone, $X = (25.4 \pm 6.2)h^{3/4}$, for the concave and straight

section, respectively. The model grounding zone width for the profile C is overestimated by approximately $10\%$ (the measured width is 2984 m ), while there is a larger discrepancy with the model grounding zone width for the concave section, which is underestimated by $18\%$ (measured width 4220 m or 4560 m using DInSAR). This is because the concave shape allows internal stresses to provide support over a longer distance (*Rabus and Lang*, 2002), leading to a wider grounding zone for a given ice thickness compared to the simple beam model.

Over a straight section of the Carlson Inlet (cross-section A) the ice thickness at point H is 1081 m, which corresponds to a modelled grounding width of 4788 m. The ice thickness here is overestimated by $30\%$ compared to the measured width of




3210 m (or 3260 m using DInSAR). This discrepancy is larger than over the straight section of the Evans Ice Steam (profile C in Figures 3 and 4), suggesting that either the ice is thinner, or the elastic modulus of the ice is lower. Areas of high stress in fast-flowing areas caused by localised high friction regions of the bed or side drag at the margins of the ice streams could

cause fracturing or alter the ice fabric making it weaker. This could potentially be why there is 10% overestimation of the elastic beam model over the Evans Ice Steam. However the ice is effectively stagnant over the Carlson Inlet, so this difference is probably due to an underestimation of ice thickness using the assumption of hydrostatic equilibrium or other unmodelled processes.

Fracturing of the ice and the reduction of its effective thickness (relative to the modelled value) can also be caused by the

motion of the ice shelf due tides (*Hulbe et al.*, 2016; *Rosier et al.*, 2017). If we compare the relationship of grounding line width of a simple elastic beam for areas of a large tidal range (>1 m) and small tidal range (<1 m) we find $X = (24.5 \pm 6.2)h^{3/4}$ and $X = (27.6 \pm 6.2)h^{3/4}$ respectively, suggesting the ice is weaker for higher tidal ranges.

Other models that include 2-D flexure of the ice shelf (*Schmeltz et al.*, 2002; *Marsh et al.*, 2014) or modelling ice as a viscoelastic material (*Wild et al.*, 2018) would provide a more accurate representation of the grounding zone. However, a more

sophisticated model is beyond the scope of the present study, and without including other factors that determine grounding zone width, we can only qualitatively asses the differences in ice thickness observed.

## 6 Conclusions

We used 7.5 years of CryoSat-2 SARIn POCA and swath data to map points F and H of the Antarctic grounding zone. We managed to obtain near-complete coverage of the grounding zones of the Siple Coast region of the Ross Ice Shelf and Filchner-

Ronne Ice Shelf. However, in lower latitude areas, further north, coverage is variable. Where the tidal range is small and swath data was the primary data source for resolving the tidal signal (e.g. the Admunsen Sea sector) we lose coverage, while in areas with a larger tidal range such as Dronning Maud land and the Larsen Ice Shelf we were able to map a significant proportion of the grounding zone. The mapped point F compared well to previous methods with a negligible bias of -0.1 km and -0.1 km and standard deviation of 1.1 km and 1.5 km between DInSAR and ICESat measurements, respectively.

The results of mapping points F and H were then used to investigate the spatial distribution of the grounding zone width, $X$, across Antarctica. This allowed us to investigate grounding zone structure across a significant fraction of the Antarctic coastline. $X$ showed a strong regional variation, and to a first approximation, the grounding line width is dependent on ice thickness. Relating our results to an elastic beam model of the grounding zone and ice shelf thickness measurements, we found $X = (25.4 \pm 0.3)h^{3/4}$, which compares well to previous studies. There was considerable scatter in the fit, however, as there are

measurement errors in both ice shelf thickness and grounding zone width, as well as un-modelled factors (such as grounding zone shape and ice rheology). For example over the Carlson Inlet and Evans Ice Stream, the simple beam model grounding line widths derived from ice shelf thickness showed significant deviations from the measured grounding line width using both CroySat-2 and DInSAR. Without the use of a more complex model of the grounding zone we could only qualitatively investigate potential factors which caused these deviations.





In areas where the grounding line has significantly retreated (e.g. the Amundsen Sea sector), the coverage was too sparse to detect any change. However, in areas where we were able to map a continuous grounding line, this method has the potential to monitor grounding line retreat and change in its structure, throughout the lifetime of the CryoSat-2 satellite.

*Data availability.*   The data sets generated during this study are available at doi TBC.

*Author contributions.*   GJD undertook the data analysis developed the methods and wrote the paper. JLB conceived the study and both authors
commented on the manuscript.

*Competing interests.*   The authors declare no competing interest

*Acknowledgements.*   This work was supported by the UK Natural Environment Research Council (NERC) grant NE/N011511/1. The European Space Agency (ESA) provided the CryoSat-2 data used for this research. We thank L. Gray for his advice in processing the CryoSat-2 swath data.





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



**Figure 2.** The dimensionless tidal amplitude ,$T_d$ for a) Dronning Maud Land, b) Filchner-Ronne Ice Shelf, c) Amery Ice Shelf, d) Antarctic Peninsula. e) Amundsen Sea Sector and f) Ross Ice Shelf.

**Figure 3.** DInSAR interference fringes over the Evans Ice Stream and the Carlson Inlet of the Filchner-Ronne Ice Shelf, along with the CryoSat-2 mapped grounding line (points F and H, solid black lines), each fringe corresponds to 2.8 cm change in height due to the motion of tides.





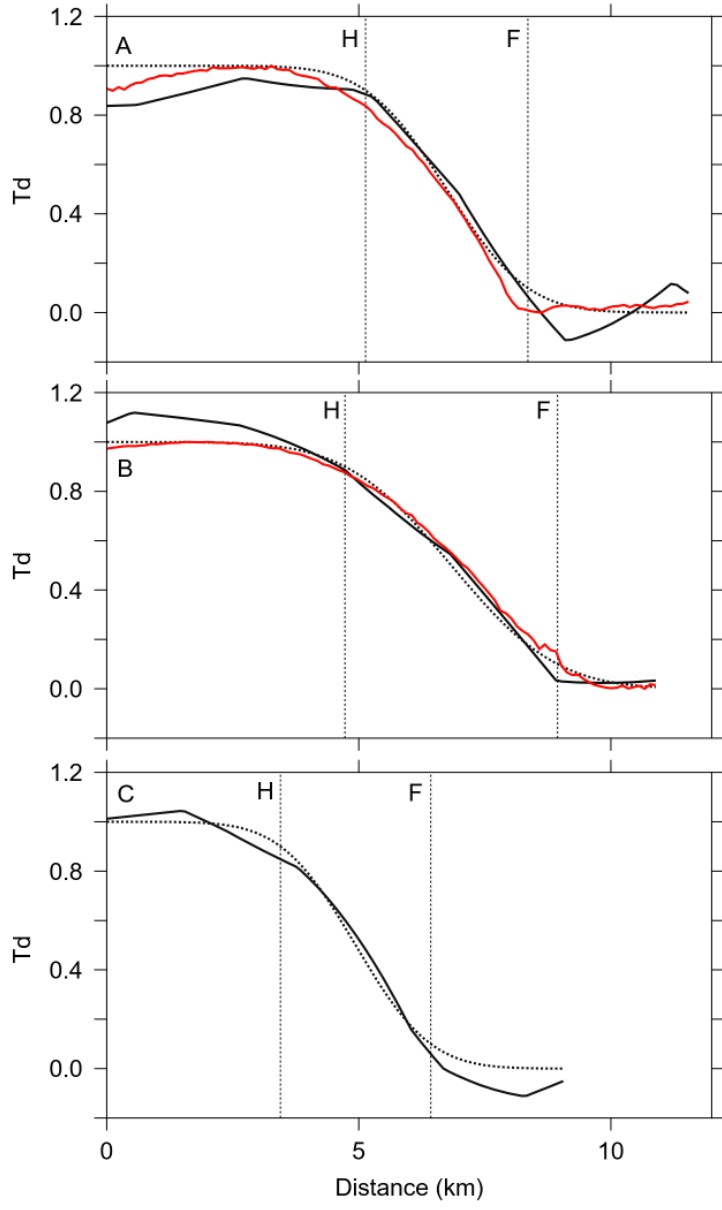

**Figure 4.** Cross-sections of $T_d$ (solid black line) with the fitted error function (dashed black line) and the normalised deformation measured from Sentinel-1 double-differenced interferograms (red line), for three cross-sections across the Carlson Inlet (A) and Evans Ice Stream (B and C) as shown in Figure 3.





**Figure 5.** Grounding line width, *W* for a) Dronning Maud Land, b) Filchner-Ronne Ice Shelf, c) Amery Ice Shelf, d) Antarctic Peninsula. e) Amundsen Sea Sector and f) Ross Ice Shelf. The background image is the ice shelf thickness (*Chuter and Bamber*, 2015) overlain on the Bedemap2 DEM (*Fretwell et al.*, 2006) .



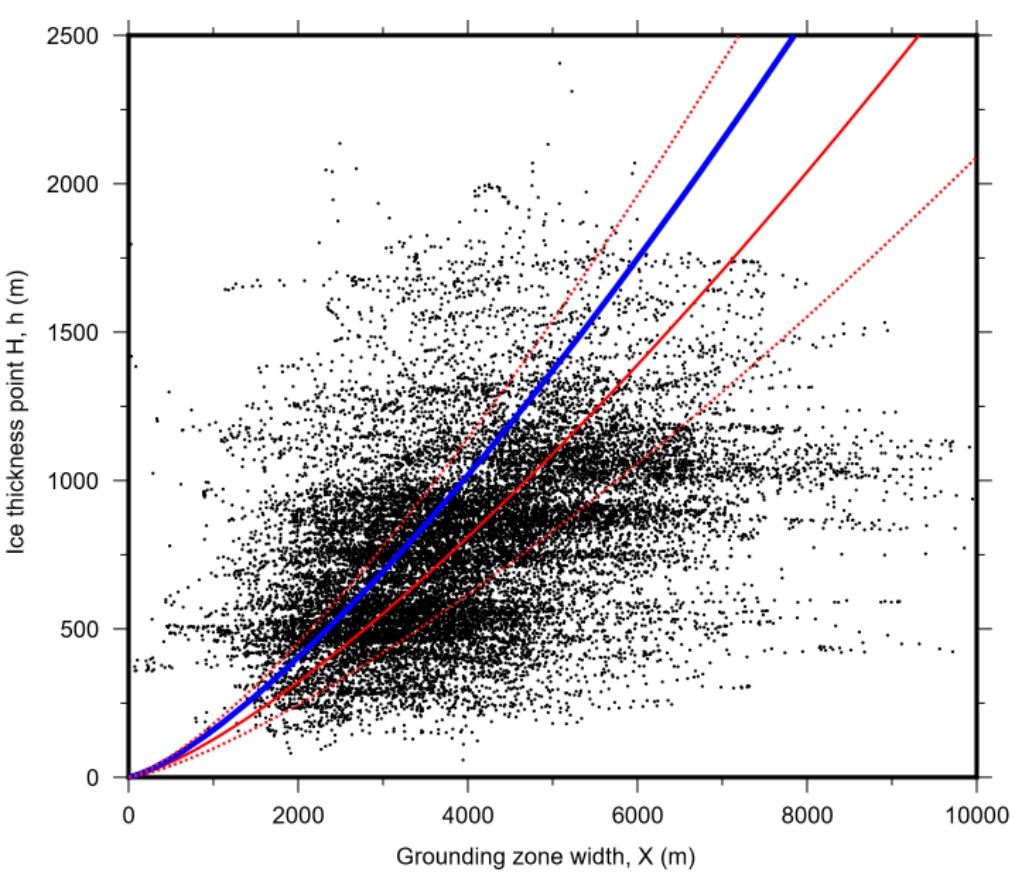

**Figure 6.** Grounding zone width (X) vs ice thickness at point H (h). The solid red line is the fit of equation $X = (26.4 \pm 6)h^{3/4}$ to the data and the dashed red lines are the uncertainty bounds, whi;e the blue line represents the estimation of *Vaughan* (1995) of $X = (25.4 \pm 6.2)h^{3/4}$.





**Figure A1.** Data coverage plot for POCA and swath data. Red, green and blue data points correspond to where we used POCA, swath or both to calculate $T_d$, respectively. The colour scales shows the data density (POCA points/km). The swath data density is scaled by 150 (the average number of swath to POCA data points) to match the POCA density for a) Dronning Maud Land, b) Filchner-Ronne Ice Shelf, c) Amery Ice Shelf, d) Antarctic Peninsula. e) Amundsen Sea Sector and f) Ross Ice Shelf.



**Figure A2.** Elevation change ($\dot{h}$) for a) Dronning Maud Land, b) Filchner-Ronne Ice Shelf, c) Amery Ice Shelf, d) Antarctic Peninsula. e) Amundsen Sea Sector and f) Ross Ice Shelf.