# Peer review of "Antarctic grounding zone characteristics from CryoSat-2"

_The Cryosphere, 2019_

## Referee Comment (RC1) · Anonymous Referee #1 · 31 Oct 2019

This paper attempts to characterize the Antarctic grounding zone using Cryosat 2 standard and swath elevation data. The authors provide 41% coverage of the larger floating ice shelves and outlet glaciers in Antarctica. The adopted grounding zone mapping methodology has already been presented in previous literature and uses an auxiliary tidal model.

I have several major criticisms both related to methodologies and conclusions:

1) The methodology adopted requires a 3 year moving window hence limiting the ability of capturing grounding line dynamic nonlinear retreats.

2) The author claims this method could potentially monitor retreat in the Amundsen Sea Embayment (ASE) . They also claim no significant changes in grounding line position

were detected over the ASE in the period 2010-2017. This contrasts with DInSAR measurements performed during the same period ( See Milillo et al 2017 for Pien Island, Milillo et al 2019 for Thwaites). In my opinion this methodology might result misleading in areas where tidal amplitudes are small and is in fact providing wrong results. The authors should also comment on the discrepancies between the aforementioned studies.

Milillo, P., Rignot, E., Mouginot, J., Scheuchl, B., Morlighem, M., Li, X., & Salzer, J. T. (2017). On the Short‐term Grounding Zone Dynamics of Pine Island Glacier, West Antarctica, Observed With COSMO‐SkyMed Interferometric Data. Geophysical Research Letters, 44(20), 10-436.

Milillo, P., Rignot, E., Rizzoli, P., Scheuchl, B., Mouginot, J., Bueso-Bello, J., & Prats-Iraola, P. (2019). Heterogeneous retreat and ice melt of Thwaites Glacier, West Antarctica. Science advances, 5(1), eaau3433.

3) The authors compare their Cryosat2 results with DInSAR Grounding line measurements, However DInSAR data have not been acquired at the same time of the Cryosat data. This important detail might result in a further misinterpretation of the results.

4) The authors use a simple elastic beam model to investigate the relationship between ice thickness and grounding zone width. The elastic model assume a fixed grounding line position whereas It has been proven in recent literature (Milillo et al 2017, Milillo et al 2019) that a simple elastic model does not explain tidally induced grounding line migrations commonly observed in nature.

For these reasons I believe this study is still immature to be published in the proposed Journal.

---

## Referee Comment (RC2) · Anonymous Referee #2 · 26 Nov 2019

This paper uses Cryosat-2 data to map Antarctic grounding lines and tries to obtain additional information about the structure of the grounding zone. A large part of the paper describes the previous work, mapping methodology and results. This in itself is not novel (in fact it is well described already in Dawson & Bamber, 2017, GRL) and while the results are extended, and reasonable agreement is shown with previous studies, the mapping does not seem to be an improvement over previous maps.

The conclusion that 'this method has the potential to monitor grounding line retreat and change in its structure', while highly valuable if true, is not supported by the results. There is no demonstration of the ability to monitor change in grounding line position or grounding zone structure over the Cryosat-2 period, nor is it clear if the confidence in these results is high enough to establish change over longer periods by comparison

with other sensors.

The title suggests the manuscript provides a method to obtain information on other grounding zone characteristics (besides position which is reasonably well known), but there are no substantial conclusions in this area. Is there any statistical significance between deviation from the grounding line width - thickness relationship and the other variables that influence ice response to tides in the grounding zone, such as plan-view curvature (i.e. concave - convex) or strain rate? These 'grounding zone characteristics' could be obtained from the results themselves, or from auxiliary datasets and could lead to interesting findings. The observation of a correlation between grounding zone width and ice thickness in-line with elastic beam theory has been shown before. In fact it is already discussed in a very similar way in Bindschadler et al., 2011.

Whilst the presentation is clear and concise, at present I feel this manuscript does not meet the standard for originality or significance required for publication in The Cryosphere. The main conclusion appears to be that Cryosat-2 tidal grounding line mapping agrees to some extent with previous studies, but does not increase accuracy, coverage or ability to monitor change. If the manuscript could be adapted to include any substantial new conclusions about grounding zone characteristics, structure or temporal or spatial change in position then it could still be a valuable contribution to the journal.

Specific comments:

L18: 'the freely'

L33: you suggest that 'DInSAR and ICESat do not have sufficient spatial or temporal coverage to monitor change across the entire grounding zone', but the method presented here also only maps 41% of the grounding zone. It is not clear what point is trying to be made and it is not clear that Cryosat-2 provides any improvement over these techniques.
L39: How can you 'characterise' a stress gradient using tidal flexure information?

L50: This is not correct. Bindschadler et al., 2011 conducted a very similar analysis to yours for the entire continent.

L84: 'closely follows' the technique in Dawson & Bamber, 2017 or is the same? If there are minor differences it would help to say what these are, or if there are no differences, say so.

L116: What is the justification for the 10km along track smoothing? Does this modify the positions significantly? What is meant by 'along track' in this context?

L138: It seems fairly arbitrary to say you would 'lose the resolution needed to map the grounding zone accurately'. What does 'accurately' mean here? How accurately is the grounding zone being mapped with the 2 km cells?

L149: A standard deviation of 1.7 km does not seem low (in comparison to your whole of Antarctica data).

Tab1: I assume 'M' means MEaSUREs and 'E' means ESA CCI. Please specify in the caption.

L172 / Fig 3: If there is no usable data from cross section C, just leave it out.

L179: Isn't the 10 km upper limit is self-imposed in the method? If you find grounding zones the full width then perhaps you need to expand the search region?

Fig 5: Bedmap-2

L185: 'Poisson's ratio is generally denoted as 'nu' not 'mu'.

L189: Rutford not Rutherford.

L199: An attempt should be made to quantify these factors.

L201-209: This is a valuable analysis. Can this be done for the whole dataset?

L219-222: How does this translate to variation in effective Young's Modulus? Is this factor actually related to change in ice properties or just to change in ability to monitor grounding zone width due to higher amplitude tides? Is there a difference between areas of semi-diurnal and diurnal tides?

L244: Cryosat & DInSAR

L246: If this is the case it needs to be shown in the paper.
* * *

---

## Referee Comment (RC3) · Anonymous Referee #3 · 4 Jan 2020

The paper present results of a newly developed tool to map the grounding zone of Antarctic ice shelves from CryoSat-2 POCA and Swath data. The method was partly introduced by the authors in 2018 using a case study and was refined and updated and applied to whole Antarctica for the present study. In total 41% of the Antarctic grounding zone and its width could be mapped in an automatic way.

The authors present in a clear and understandable way the method and compare the findings against independent grounding line data sets which were mapped using DInSar methods. The standard deviation to those datasets is around 1km with regional differences. Additional the authors compare their results directly with cross section of DInsar interferograms from Sentinel 1 and can clearly show how well both methods match but also explain differences and shortcomings of their method.

[Figure]

In the last section they apply an elastic beam model to find a relation between ice thickness and grounding zone width with similar findings as Bindschadler (2011).

The paper is well written, figures are clear and of high quality. The scientific outcome is of interest to the community, at least to my opinion, as it provides another independent data set of the grounding line and grounding zone width which is derived from Altimetry alone.

I would like to thank the authors for this excellent work as the pre-processing already incorporates a full retracking of CryoSat-2 SARin data, the estimation of the POCA and a full interferometric swath processing. This dataset is then explored in a new way to derive grounding line and grounding zone width.

I do have some minor comments and questions.

1. Can you please argue why you selected a 3-year moving window to estimate 6-yearly measurements per grid cell which were then averaged instead of using the full time series or 5-year moving windows. I could imagine that with more data points per grid cell more tidal states are covered which might allow you better results in areas with low tidal signals or sparse coverage. I can understand the argument with GL retreat however you mentioned that your approach was not able to detect a retreat in the Amundsen sector.

2. Please explain in more detail which criteria you used for the selection of SWATH and POCA data. I don't see any coverage of SWATH across the shelf ice, which makes sense as the SWATH shouldn't give useful information in flat terrain. However, Gourmelen et. al. showed some good results across Dotson. Is it possible to use Swath in the vicinity of Dotson as well in your study?

3. Please include in your validation against other grounding line data sets the ASAID dataset (Bindschadler, 2011) ASAID provides also the F and H lines and it would be a valuable information how much they differ and if you can see if and where H and F

shows better agreement. Maybe you find some systematic difference.

4. Hogg et. al. (2017) mapped the grounding line from CryoSat-2 data as well. They used a different technique (break in slope) using only POCA data. Can you please show the differences to your data set and as reference to the ASAID one. It would be really interesting to see how much the additional use of Swath data and your new approach differs. Maybe in future one can find a combined approach to overcome shortcomings e.g. your approach has difficulties in areas of low tidal signal.

5. Did you use a reference elevation model (REMA or global Tandem-X) to subtract topographic phase from your interferograms prior forming the DInSar interferogram? This might help to get rid of phase wrapping and to get a clearer picture in areas where you were not able to unwrap the DInSar phase (cross section C in Fig. 3, 4).

5. Your are following the method of Bindschadler et. al. 2011 to estimate a relation between width and ice thickness. Can you please apply the fit to different regiones to see if you can reduce the spread in cases of low measurement error.

Can you please derive your best fit using another Young modulus to show the influence of E. e.g. Rack et. al. 2017 used 1.5 Gpa to analyse the tidal flexure in the grounding zone and where able to account for horizontal motion in DInSar derived grounding line position. Whereas Wild et. al. 2019 found 1.0 +/- 0.56 GPa as best fit to tiltmeter measurements and a numerical model.

Figures:

Please note which grounding line you used in the figures 1,2, A1 and A2

Fig 1 and A2: Please change the colour scale. Red-Green blind people can't see anything.

Fig 3: Why did you select cross section C in Figure 3 as validation against DInSar? It would be also worth to show a second DInSar pair from a different tidal state, to illustrate how much the width of the fringe belt can vary. Maybe you can include the F

and H line of the ASAID data set as well.

Fig 6: Please double check the number and citation and the position of the blue line. Bindschadler (2011) derived 22.2 +/- 6.2 referring to values estimated by Vaughan (1995).

Typo: Please double check the numbers given for X in line 191 and 204 and Fig 6.

---

## Referee Comment (RC4) · Anonymous Referee #4 · 23 Mar 2020

In their TCD manuscript "Antarctic grounding zone characteristics from CryoSat-2" Dawson and Bamber employed CryoSat-2 data to map 41% of the main floating ice shelves and outlet glaciers of Antarctica. The used method closely follows the one described by Dawson and Bamber (2017) but uses 7.5 years of Cryosat-2 data and is applied to the whole of Antarctica. In contrast to their previous study the authors estimate the width of the grounding zone by fitting an error function to their CryoSat-2 estimate and compare their results with grounding zone estimates from Sentinel-1 DInSAR.

General remark:

Overall I find the manuscript is well written and interesting to read. I like the way how CryoSat-2 data is employed here as the proposed method is much more sophisticated

than previous break-in-slope assumptions of the grounding line. However, considering the limitations of the method it is difficult to judge where the results are trustworthy and where not. I therefore suggest to include a reliability map which utilizes the combined effect of tidal range and data coverage. This should result in reasonable results at high latitudes – i.e. regions which are only sparsely covered by grounding line estimates from DInSAR due to orbital constrains. To strengthen the study I would also put more emphasizes on the latter point which should be mentioned in the abstract and conclusion. Further, I encountered several flawless mistakes which need to be corrected and are partly listed in the following. Please be consistent with the terms "grounding line" and "grounding zone".

Specific comments:

Line 18: typo, "thefreely".

Line 23: include "in" before grounding line location.

Line 26: remove "of".

Line 27: I presume you mean satellite remote sensing here?

Line 29-31: maybe you could already state here that the term "grounding line" refers to point F throughout the manuscript.

Line 34: what is meant by "entire grounding zone"? Not clear.

Line 111-112: are you really referring to the grounding line (i.e point F) here? Please clarify.

Line 114: I am not sure what is meant by grounding line width? Are you referring to the grounding zone width here?

Line 120: 41% relative to what? Please state which ice shelves and outlet glaciers are defined as "main", otherwise this number is worthless. Maybe it is more appropriate to state that you were able to map 31% of the grounding zone surrounding Antarctica (at

least according to your Table 1). This also applies for the abstract.

Line 121-122: I think this is a very important point, as these are the critical areas for DInSAR estimates due to orbital constrains. Here only few coherent left looking acquisitions are available from TerraSAR-X and RADARSAT drawing a rather incomplete picture of the grounding zone. Further, break-in-slope estimates are far off due to gentle slopes in the area. It would certainly strengthen the manuscript if more emphasizes would be on this point.

Line 131: maybe you could also cite Gourmelen et al., 2017 here as their study is also based on CryoSat-2.

Line 142: I am wondering why the results are not compared to the ones from Bindschadler et al., 2011?

Line 181: I am not sure what you mean by grounding line width? Width of the grounding zone? If so, please change here and elsewhere.

Line 201-209: this could potentially be shown in Figure 4.

Line 211: grounding zone?

Line 211: are you sure you are referring to ice thickness here?

Line 220: include "to" before "tides".

Line 224-226: true, therefore I find the section title "Grounding zone structure" a little bit misleading.

Figure 1: which grounding line is shown here? This needs to be cited in the caption as it is certainly not the one derived in this study.

Figure 4: please state in the caption that you were not able to unwrap the fringe belt at the location of profile C.

Figure 5: "Grounding line width, W" has never be mentioned in the text. I am not

really convinced about the information content of this Figure and would rather move it to the appendix. Instead I would include a reliability map into the main manuscript as mentioned in my general remark.

Additional References:

Gourmelen, N., Goldberg, D. N., Snow, K., Henley, S. F., Bingham, R. G., Kimura, S., . . . van de Berg, W. J. (2017). channelized melting drives thinning under a rapidly melting Antarctic ice shelf. Geophysical Research Letters, 44, 9796– 9804. https://doi.org/10.1002/2017GL074929

---

## Author Comment (AC1) · 17 Apr 2020

**Dear Editors and Reviewers**

Thank you for your time in reviewing our submission. We are grateful for the opportunity to respond to the reviewer's comments, which have proven extremely useful in revising this manuscript. We have now revised the manuscript offering additional information and context.

Below you will find a detailed response to each comment presented by the reviewers and how these changes are incorporated in the revised manuscript.

Yours sincerely

**Geoffrey Dawson**

**REVIEWER 1**

1) The methodology adopted requires a 3 year moving window hence limiting the ability of capturing grounding line dynamic nonlinear retreats.

The main purpose of this study was to map the grounding zone and not to capture grounding line dynamic nonlinear retreats. However, it is correct that the method presented here cannot capture these events, and we have now included this as a limitation (line 100).

2) The author claims this method could potentially monitor retreat in the Amundsen Sea mbayment (ASE). They also claim no significant changes in grounding line position were detected over the ASE in the period 2010-2017. This contrasts with DInSAR measurements performed during the same period (See Milillo et al 2017 for Pien Island, Milillo et al 2019 for Thwaites). In my opinion this methodology might result misleading in areas where tidal amplitudes are small and is in fact providing wrong results. The authors should also comment on the discrepancies between the aforementioned studies.

We are unsure where this comment originates from. We state clearly in the manuscript that we could not map the grounding line in the ASE area, and hence we have not made any claims about grounding line retreat.

In line 126, we clearly articulate that we could not map the grounding zone in these areas:

"In the high sloping, low tidal range (0.8 m - 1 m) Amundsen Sea sector and the Amery Ice Shelf, we could not map a continuous grounding line."

The only place where the reviewer could have misinterpreted this is in line 245:

"In areas where the grounding line has significantly retreated (e.g. the Amundsen Sea sector), the coverage was too sparse to detect any change."

Which to avoid any further confusion, we have changed to:

"In areas where the grounding line has significantly retreated (e.g. the Amundsen Sea sector), we did not obtain sufficient coverage to map the grounding."

3) The authors compare their Cryosat2 results with DInSAR Grounding line measurements, However DInSAR data have not been acquired at the same time of the Cryosatdata. This important detail might result in a further misinterpretation of the results.

The area that we compared our results to, is known to be a stable area with no observed grounding line migration. Therefore, we are confident that comparing the results one year apart will lead to a negligible difference in grounding line location.

4) The authors use a simple elastic beam model to investigate the relationship between ice thickness and grounding zone width. The elastic model assume a fixed grounding line position whereas It has been proven in recent literature (Milillo et al 2017, Milillo et al 2019) that a simple elastic model does not explain tidally induced grounding line migrations commonly observed in nature.

We agree that a simple elastic model does not explain tidally induced grounding line migrations commonly observed in nature. It has also been proven in the literature that the simple elastic model does not capture the ice rheology, the motion of the ice sheet, grounding line geometry, or effects of tidal range. However, as we are comparing measurements of the grounding zone width over a large portion of the grounding zone of Antarctica, it was not feasible within the scope of this study to use a more complex model, as we have highlighted in this manuscript. We have now included tidally induced grounding migrations as another limitation of the model (line 243).

For these reasons I believe this study is still immature to be published in the proposed Journal.

We are disappointed in this review, in particular, comment 2, as this clearly shows the reviewer has not thoroughly read the manuscript. The other 3 criticisms are either beyond the scope of the study or only require minor changes which we have addressed in the relevant sections (as stated above).

**REVIEWER 2**

This paper uses Cryosat-2 data to map Antarctic grounding lines and tries to obtain additional information about the structure of the grounding zone. A large part of the paper describes the previous work, mapping methodology and results. This in itself is not novel (in fact it is well described already in Dawson & Bamber, 2017, GRL) and while the results are extended, and reasonable agreement is shown with previous studies, the mapping does not seem to be an improvement over previous maps.

While this method is not shown to be an improvement over previous studies, it can provide additional coverage that can contribute to the overall spatio-temporal mapping of the grounding zone. To support this conclusion, we have added another section titled (5. Coverage comparison with other methods). This section describes how the coverage of the CryoSat-2 data relates to other products. We show that there are several sections along the Filchner-Ronne Ice shelf that have previously only been mapped using break-in-slope methods. The previous grounding point F measurements that coincide with the new CryoSat-2 show no significant deviations. And while this does not show any spatial change in position, the results still confirm the location and stability of the mapped grounding zones.

The conclusion that 'this method has the potential to monitor grounding line retreat and change in its structure', while highly valuable if true, is not supported by the results. There is no demonstration of the ability to monitor change in grounding line position or grounding zone structure over the Cryosat-2 period, nor is it clear if the confidence in these results is high enough to establish change over longer periods by comparison

**We agree that our results do not directly show the potential to monitor grounding line retreat, and therefore, we have removed this statement.**

The title suggests the manuscript provides a method to obtain information on other grounding zone characteristics (besides position which is reasonably well known), but there are no substantial conclusions in this area. Is there any statistical significance between deviation from the grounding line width - thickness relationship and the other variables that influence ice response to tides in the grounding zone, such as plan-view curvature (i.e. concave - convex) or strain rate? These 'grounding zone characteristics' could be obtained from the results themselves, or from auxiliary datasets and could lead to interesting findings. The observation of a correlation between grounding zone width

and ice thickness in-line with elastic beam theory has been shown before. In fact it is already discussed in a very similar way in Bindschadler et al., 2011.

Unfortunately, the data was too noisy for any statistically significant results to derive any firm conclusions. As a result, we have now removed lines 201-227 discussing the grounding zone structure, as we cannot reliably discuss any inferences about grounding zone characteristics. We have also altered the title of the manuscript to 'Measuring the location and width of the Antarctic grounding zone using CryoSat-2' as this changes the focus to mapping the grounding zone. However, the analysis using the elastic beam theory still provides valuable results as we have now calculated an effective Young's modulus of ice, which agrees well with previous methods.

Whilst the presentation is clear and concise, at present I feel this manuscript does not meet the standard for originality or significance required for publication in The Cryosphere. The main conclusion appears to be that Cryosat-2 tidal grounding line mapping agrees to some extent with previous studies, but does not increase accuracy, coverage or ability to monitor change. If the manuscript could be adapted to include any substantial new conclusions about grounding zone characteristics, structure or temporal or spatial change in position then it could still be a valuable contribution to the journal.

We thank the reviewer for their detailed comments and discussing weaknesses in the manuscript. We have now made significant alterations to the manuscript in response to these comments. We have now included a new section (5. Coverage comparison with other methods) detailing the additional coverage provided by this new dataset. This new section discusses the grounding line coverage provided by this method and other datasets, and while the mapped grounding zone mostly overlaps with previous methods, it still provides additional coverage, both spatiality and temporally.

As mentioned in previous responses, we could not perform any further analysis of the grounding zone structure. And in response to the reviewer's comments, we have removed the discussion on grounding zone structure beyond the simple elastic model as it did not provide any significant conclusions. The revised manuscript now highlights the valuable results of this research, while removing any inconclusive discussion present in the earlier version.

**Specific comments:**

L18: 'the freely'

**corrected**

L33: you suggest that 'DINSAR and ICESat do not have sufficient spatial or temporal coverage to monitor change across the entire grounding zone', but the method presented here also only maps 41% of the grounding zone. It is not clear what point is trying to be made and it is not clear that Cryosat-2 provides any improvement over these techniques.

CryoSat-2 data will not necessarily offer any improvements over DInSAR or laser altimetry techniques. However, despite only mapping 31% of the grounding zone, we do map point F for some sections of the grounding zone that have not been mapped in previous research (as this has been added to Section 5). See above for further discussions on this.

L39: How can you 'characterise' a stress gradient using tidal flexure information?

We agree this is ambiguous so have altered the statement to 'determine thickness and rheology across the grounding zone'

L50: This is not correct. Bindschadler et al., 2011 conducted a very similar analysis to yours for the entire continent.

We have altered to say 'mostly', Bindschadler et al., 2011 did investigate the grounding zone width over the entire continent. However, ours is the first study to compare grounding zone width calculated with tidal flexure information over a large percentage of the grounding zone.

L84: 'closely follows' the technique in Dawson & Bamber, 2017 or is the same? If there are minor differences it would help to say what these are, or if there are no differences, say so.

We have now clearly outlined in the revised manuscript where the methodology differs. In line 94, we state that we used an additional linear surface elevation rate, and in line 103 we explain that we used the moving window approach as we used 7.5 years of data instead of 3 years (as in the previous study).

L116: What is the justification for the 10km along track smoothing? Does this modify the positions significantly? What is meant by 'along track' in this context?

We have now changed along-track to along-line. The 10km smoothing does not modify the position of the grounding zone significantly and removes noise related to incorrectly mapping points F or H. This is now mentioned in line 133.

L138: It seems fairly arbitrary to say you would 'lose the resolution needed to map the grounding zone accurately'. What does 'accurately' mean here? How accurately is the grounding zone being mapped with the 2 km cells?

By 'accurately', we meant to be able to map with sufficient precision so that it is comparable to other methods and detect any changes in the grounding line position. We have now changed the wording (line 155).

This is effectively an arbitrary choice; however, we needed the data to be of comparable precision to previous methods. As we demonstrate in section 4.2, the CryoSat-2 grounding lines can be mapped with a standard deviation of 1.5 km compared to previous methods using a 2 km cell. This is similar to comparing different methods; for example, the DInSAR grounding line differs from the ICESat-1 grounding line by a standard deviation of 1.1 km. Therefore, using a larger cell size would lower the precision of the grounding zone map, and the results would not be comparable to other methods.

L149: A standard deviation of 1.7 km does not seem low (in comparison to your whole of Antarctica data). Tab1: I assume 'M' means MEaSUREs and 'E' means ESA CCI. Please specify in the caption.

You are correct, this has now been added

L172 / Fig 3: If there is no usable data from cross section C, just leave it out.

Agree

L179: Isn't the 10 km upper limit is self-imposed in the method? If you find grounding zones the full width then perhaps you need to expand the search region?

We have removed this sentence to avoid any confusion and added a sentence (line 129) clarifying that all mapped regions were below the 10 km limit.

**Fig 5: Bedmap-2**

Amended

L185: 'Poisson's ratio is generally denoted as 'nu' not 'mu'.

Agree and amended

L189: Rutford not Rutherford.

**Agree and amended**

L199: An attempt should be made to quantify these factors.

As mentioned previously the data is too scattered to attempt to quantify these factors.

L201-209: This is a valuable analysis. Can this be done for the whole dataset?

The data is too scattered to apply to the whole dataset and as a response to your earlier comments, we have removed this from the manuscript.

L219-222: How does this translate to variation in effective Young's Modulus? Is this factor actually related to change in ice properties or just to change in ability to monitor grounding zone width due to higher amplitude tides? Is there a difference between areas of semi-diurnal and diurnal tides?

We could not observe a difference between semi-diurnal and diurnal tides. As mentioned previously we have removed this section as without other statistically significant results the conclusion that this result can be believed is very weak.

L244: Cryosat & DInSAR

This section has been removed

L246: If this is the case it needs to be shown in the paper

This section has been removed

**REVIEWER 3**

The paper present results of a newly developed tool to map the grounding zone of Antarctic ice shelves from CryoSat-2 POCA and Swath data. The method was partly introduced by the authors in 2018 using a case study and was refined and updated and applied to whole Antarctica for the present study. In total 41% of the Antarctic grounding zone and its width could be mapped in an automatic way. The authors present in a clear and understandable way the method and compare the findings against independent grounding line data sets which were mapped using DInSar methodes. The standard deviation to those datasets is around 1km with regional differences. Additional the authors compare their results directly with cross section of DInsar interferograms from Sentinel 1 and can clearly show how well both methods match but also explain differences and shortcomings of their method.

In the last section they apply an elastic beam model to find a relation between ice thickness and grounding zone width with similar findings as Bindschadler (2011). The paper is well written, figures are clear and of high quality. The scientific outcome is of interest to the community, at least to my opinion, as it provides another independent data set of the grounding line and grounding zone width which is derived from Altimetry alone. I would like to thank the authors for this excellent work as the pre-processing already incorporates a full retracking of CryoSat-2 SARin data, the estimation of the POCA and a full interferometric swath processing. This dataset is then explored in a new way to derive grounding line and grounding zone width.

I do have some minor comments and questions.

1. Can you please argue why you selected a 3-year moving window to estimate 6- yearly measurements per grid cell which were then averaged instead of using the full time series or 5-year moving windows. I could imagine that with more data points per grid cell more tidal states are covered which might allow you better results in areas with low tidal signals or sparse coverage. I can

understand the argument with GL retreat however you mentioned that your approach was not able to detect a retreat in the Amundsen sector.

We have tested the full time series and the 5-year moving window per grid cell, and neither improved our results. As we mention in line 103, we used a 3-year moving window to account for any non-linear elevation change over the time period. And even though we were not able to obtain results over the grounding zones where this has occurred e.g. the Amundsen sector, this is still a good method to account for any variation like this, that may occur in areas where we have better coverage.

2. Please explain in more detail which criteria you used for the selection of SWATH and POCA data. I don't see any coverage of SWATH across the shelf ice, which makes sense as the SWATH shouldn't give useful information in flat terrain. However, Gourmelen et. al. showed some good results across Dotson. Is it possible to use Swath in the vicinity of Dotson as well in your study?

We used both POCA and swath data throughout the study where available in accordance to the processing parameters given in lines 74-75. There is also coverage of swath data over the ice shelves, as shown in Figure 1, where the ice shelves show a combination of POCA and swath data being used. The swath data improves coverage over the ice shelves particularly where there are crevassed regions. The majority of the data used is POCA where parts of the ice shelves are flat. However, we agree that we have not added sufficient detail regarding the use of POCA and swath data, particularly over the ice shelves and this has been added to line 84.

3. Please include in your validation against other grounding line data sets the ASAID dataset (Bindschadler, 2011) ASAID provides also the F and H lines and it would be a valuable information how much they differ and if you can see if and where H and F shows better agreement. Maybe you find some systematic difference.

We have not used the ASAID dataset for comparison as we wanted to compare methods that measured points F and H from tidal flexure information not the break-in-slope methods. However, this is a valuable dataset and we have now included it in section 5, discussing the location of the grounding lines from various methods.

4. Hogg et. al. (2017) mapped the grounding line from CryoSat-2 data as well. They used a different technique (break in slope) using only POCA data. Can you please show the differences to your data set and as reference to the ASAID one. It would be really interesting to see how much the additional use of Swath data and your new approach differs. Maybe in future one can find a combined approach to overcome shortcomings e.g. your approach has difficulties in areas of low tidal signal.

The Hogg 2017 grounding line measures the break-in-slope and as our method measures the limit of tidal flexure, they are effective measuring different things, thus combing the two would be difficult. Keeping them as separate but complementary products would be advisable.

5. Did you use a reference elevation model (REMA or global Tandem-X) to subtract topographic phase from your interferograms prior forming the DInSar interferogram? This might help to get rid of phase wrapping and to get a clearer picture in areas where you were not able to unwrap the DInSar phase (cross section C in Fig. 3, 4).

**We used REMA in our processing, this information has been added to the manuscript (see line 179).**

5. Your are following the method of Bindschadler et. al. 2011 to estimate a relation between width and ice thickness. Can you please apply the fit to different regiones to see if you can reduce the spread in cases of low measurement error. Can you please derive your best fit using another Young modulus to show the influence of E. e.g. Rack et. al. 2017 used 1.5 Gpa to analyse the tidal flexure in the grounding zone and where able to account for horizontal motion in DInSar derived grounding line position. Whereas Wild et. al. 2019 found 1.0 +/- 0.56 GPa as best fit to tiltmeter measurements and a numerical model.

We did not derive our results with any Young's modulus as this is a parameter which is calculated within the fit, and can be derived, if we use assumptions for seawater density and the Poisson ratio of ice. We have now modified this section to explain this more explicitly. We also derived a Young's modulus from this work with  $E=1.4 \pm 0.9$  GPa, which agrees well with previous studies. As mentioned in the response to reviewer 2, the data was too noisy to make any further comparisons, for example, we did not see statistically significant results between different regions.

**Figures:**

Please note which grounding line you used in the figures 1,2, A1 and A2 Fig 1 and A2: Please change the colour scale. Red-Green blind people can't see anything.

We have added the grounding line used in the Figure caption. We found it extremely difficult to have a suitable colour scale in Figures 1 and A2 as we include intensities as well as three different scales. Instead we have added a new Figure (A3) which is a replication of A2 but with a colour-blind safe scale without intensity values.

Fig 3: Why did you select cross section C in Figure 3 as validation against DInSar? It would be also worth to show a second DInSar pair from a different tidal state, to illustrate how much the width of the fringe belt can vary. Maybe you can include the F and H line of the ASAID data set as well.

We have removed cross section C. We used this DInSAR data as the difference in tides between the two scenes is 1.1m and this is similar to the average displacement due to tides over the Ronne Ice Shelf. As the CryoSat-2 method effectively measures the average of point F and H as it samples the grounding zone over a long period of time, therefore using this image is a fair comparison. We have added a sentence in line 186 to discuss this point. We agree that investigating point F and H in relation to different tidal amplitudes would be an interesting study, however this would not provide any further validation for out CryoSat-2 data.

Fig 6: Please double check the number and citation and the position of the blue line. Bindschadler (2011) derived 22.2 +/- 6.2 referring to values estimated by Vaughan (1995). Typo: Please double check the numbers given for X in line 191 and 204 and Fig 6.

Checked and altered

**REVIEWER 4**

In their TCD manuscript "Antarctic grounding zone characteristics from CryoSat-2" Dawson and Bamber employed CryoSat-2 data to map 41% of the main floating ice shelves and outlet glaciers of Antarctica. The used method closely follows the one described by Dawson and Bamber (2017) but uses 7.5 years of Cryosat-2 data and is applied to the whole of Antarctica. In contrast to their previous study the authorsestimate the width of the grounding zone by fitting an error function to their CryoSat2 estimate and compare their results with grounding zone estimates from Sentinel-1DInSAR.

General remark:

Overall I find the manuscript is well written and interesting to read. I like the way how CryoSat-2 data is employed here as the proposed method is much more sophisticated than previous break-inslope assumptions of the grounding line. However, considering the limitations of the method it is difficult to judge where the results are trustworthy and where not. I therefore suggest to include a reliability map which utilizes the combined effect of tidal range and data coverage. This should result in reasonable results at high latitudes – i.e. regions which are only sparsely covered by grounding line estimates from DInSAR due to orbital constrains.

Although we have not included a reliability map made from information about the data coverage and tidal range, we have now added a map of the standard deviation of  $T_d$  (calculated from the yearly measurements). This map can act as a guide to the reliability of the data. We have included a description in lines 113-115 and discussed the results in lines 169-170. We find the standard deviation is lowest in the high tidal range areas and high latitude areas while it increases to the low tidal range and low latitude areas. These results closely match what we observe in comparison with DInSAR and ICESat-1 measurements.

To strengthen the study I would also put more emphasizes on the latter point which should be mentioned in the abstract and conclusion.

We have now included this point in the abstract and conclusion and has been discussed in the new section (5. Coverage comparison with other methods)

Further, I encountered several flawless mistakes which need to be corrected and are partly listed in the following. Please be consistent with the terms "grounding line" and "grounding zone".

Specific comments:

Line 18: typo, "thefreely".

Amended

Line 23: include "in" before grounding line location.

Amended

Line 26: remove "of".

Amended

Line 27: I presume you mean satellite remote sensing here?

**Yes and amended**

Line 29-31: maybe you could already state here that the term "grounding line" refers to point F throughout the manuscript.

**Agree and amended**

Line 34: what is meant by "entire grounding zone"? Not clear.

Changed to 'across all the Antarctic grounding zone'

Line 111-112: are you really referring to the grounding line (i.e point F) here? Please clarify.

Yes, and based on your previous comment we have now explicitly mentioned the grounding line refers to point F.

Line 114: I am not sure what is meant by grounding line width? Are you referring to the grounding zone width here?

**Agree and amended**

Line 120: 41% relative to what? Please state which ice shelves and outlet glaciers are defined as "main", otherwise this number is worthless. Maybe it is more appropriate to state that you were

able to map 31% of the grounding zone surrounding Antarctica (at least according to your Table 1). This also applies for the abstract.

**Agree and amended**

Line 121-122: I think this is a very important point, as these are the critical areas for DInSAR estimates due to orbital constrains. Here only few coherent left looking acquisitions are available from TerraSAR-X and RADARSAT drawing a rather incomplete picture of the grounding zone. Further, break-in-slope estimates are far off due to gentle slopes in the area. It would certainly strengthen the manuscript if more emphasizes would be on this point.

We have added a new section that highlights this point titled '5. Coverage comparison with other methods'. In this section, we describe in detail the grounding line coverage provided by this method compared to other datasets, and we show there are some areas in the high latitude areas that have not been mapped using DInSAR.

Line 131: maybe you could also cite Gourmelen et al., 2017 here as their study is also based on CryoSat-2.

Yes

Line 142: I am wondering why the results are not compared to the ones from Bindschadler et al., 2011?

We did not want to compare these results directly with Bindschadler et al., 2011 as these results are from break-in-slope methods. However, we have added a comparison with this dataset in section 5

Line 181: I am not sure what you mean by grounding line width? Width of the grounding zone? If so, please change here and elsewhere.

**Agree and amended**

Line 201-209: this could potentially be shown in Figure 4.

We have now removed this section in response to reviewer 2.

Line 211: grounding zone?

We have now removed this section in response to reviewer 2.

Line 211: are you sure you are referring to ice thickness here?

We have now removed this section in response to reviewer 2.

Line 220: include "to" before "tides".

We have now removed this section in response to reviewer 2.

Line 224-226: true, therefore I find the section title "Grounding zone structure" a little bit misleading.

We agree and we have now changed the title to be 'Grounding zone width'

Figure 1: which grounding line is shown here? This needs to be cited in the caption as it is certainly not the one derived in this study.

**Agree and amended**

Figure 4: please state in the caption that you were not able to unwrap the fringe belt at the location of profile C.

**We have now removed profile C from the figure.**

Figure 5: "Grounding line width, W" has never be mentioned in the text. I am not really convinced about the information content of this Figure and would rather move it to the appendix. Instead I would include a reliability map into the main manuscript as mentioned in my general remark.

The map grounding zone width is discussed in section 6 and provides valuable information about the spatial distribution of the width of the grounding zone, and therefore should be left in the main text. However, in response to the earlier comment, we have included a map that gives a measure of the reliability of the data (this has been included in the appendix).

---

## Author Comment (AC2) · 17 Apr 2020

Please find our combined response to the reviewers in the Supplement.

Please also note the supplement to this comment:
https://www.the-cryosphere-discuss.net/tc-2019-196/tc-2019-196-AC2-supplement.pdf
* * *